# Deep Connectomics Networks: Neural Network Architectures Inspired by Neuronal Networks

**Nicholas Roberts**
Carnegie Mellon University
nick11roberts@cmu.edu

**Dian Ang Yap**
Stanford University
dayap@stanford.edu

**Vinay Uday Prabhu**
UnifyID AI Labs
vinay@unify.id

## Abstract

The interplay between inter-neuronal network topology and cognition has been studied deeply by connectomics researchers and network scientists, which is crucial towards understanding the remarkable efficacy of biological neural networks. Curiously, the deep learning revolution that revived neural networks has not paid much attention to topological aspects. The architectures of deep neural networks (DNNs) do not resemble their biological counterparts in the topological sense. We bridge this gap by presenting initial results of *Deep Connectomics Networks* (DCNs) as DNNs with topologies inspired by real-world neuronal networks. We show high classification accuracy obtained by DCNs whose architecture was inspired by the biological neuronal networks of *C. Elegans* and the mouse visual cortex.

## 1   Introduction

Recent advancements in neural network models have emerged through research in network architectures (He et al., 2016; Krizhevsky et al., 2012; Simonyan & Zisserman, 2014), optimization (Kingma & Ba, 2014; Liu et al., 2019; Luo et al., 2019), and generalization techniques (Srivastava et al., 2014; Ioffe & Szegedy, 2015; Zhang et al., 2019), with convolutional layers inspired from receptive fields and functional architecture in cats' visual cortex. However, the field of deep neural networks, with all its neuro-biologically inspired building blocks, has mostly left the topology story out.[1] Curiously, in the Cambrian explosion of neural network architectures in the post-AlexNet era, none seem to be inspired by the ideas prevalent in the domain of brain connectomics.[2]

The field of neuroscience was drawn into the network sciences when (Watts & Strogatz, 1998) introduced the small-world network model, an example of which is the neuronal network of the nematode Caenorhabditis elegans (*C. Elegans*)[3]. This idea was a sharp departure from the literature at the time as it considered a network model which was neither completely regular nor completely random - a model of complex networks.

Complex networks with small world topology (Watts & Strogatz, 1998) serve as an attractive model for the organization of brain anatomical and functional networks because a small-world topology can support both segregated/specialized and distributed/integrated information processing (Bassett & Bullmore, 2006). Interestingly, while applications of complex networks based modeling have been well explored by the neuroscience community, they have been largely unexplored by the machine learning community as an avenue for designing and understanding deep learning architectures.

---

[1] These sentences appear verbatim in (LeCun et al., 2015):   *"The convolutional and pooling layers in ConvNets are directly inspired by the classic notions of simple cells and complex cells in visual neuroscience, and the overall architecture is reminiscent of the LGN-V1-V2-V4-IT hierarchy in the visual cortex ventral pathway."*

[2] The growing neural-network zoo: http://www.asimovinstitute.org/neural-network-zoo/.

[3] As of 2019, *C. Elegans* is the only organism to have its connectome (neuronal "wiring diagram") completed.

Bridging the gap between neural connectomics (Fornito et al., 2016) and deep learning, we propose initial findings of designing neural network wiring based on connectomic structures as an intersection between network sciences, neuroscience (Bassett & Sporns, 2017) and deep learning, and test our findings on image classification.

## 2 Related Work

**Small-World Networks.** By rewiring regular networks to introduce higher entropy and disorder, Watts and Strogatz proposed small-world networks with high clustering and small average path length. The model is analogous to six degrees of separation in the small-world phenomenon (Watts & Strogatz, 1998). Small-world networks are present in *C. Elegans*'s connectome, power grid networks and protein interactions (Telesford et al., 2011).

**Small-world models in neuroscience.** Human brain structural and functional networks follow small-world configuration and this small-world model captures individual cognition and exhibits physiological basis (Liao et al., 2017). In the field of development psychology, literature shows that small-world modules and hubs are present during the mid-gestation period, and early brain network topology can predict later behavioral and cognitive performance (Zhao et al., 2018; Watts & Strogatz, 1998).

Erdos-Renyi (ER) (Erdős & Rényi, 1960), Barabasi-Albert (BA) (Albert & Barabási, 2002), and Watts-Strogatz (WS) (Watts & Strogatz, 1998) models, Xie et. al. applied these graphs for image classification and showed that randomly wired neural networks achieve competitive accuracy on the ImageNet benchmark(Xie et al., 2019).

## 3 Methods and Experiments

Successes of ResNets (He et al., 2016) and DenseNets (Huang et al., 2017) in their performance as the first convolutional neural network (CNNs) that surpasses human-level performance on ImageNet were largely attributed to creative wiring patterns, with skip connections between multiple localized convolutional layers that is analogous to long-range connections across dense localized clusters, akin to small-world networks in neuronal networks. Inspired by small-world structures in deep CNNs, we construct DCNs based on biological neuronal network patterns, and determine their effectiveness in image classification[4].

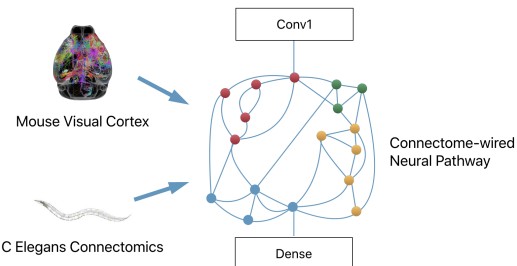

Figure 1: Wired graphs adopt connectomics structure from mouse and *C. Elegans* connectomes. Colored nodes indicate localized clusters in DCNs with small-world structures.

### 3.1 Construction of Directed Acyclic Graphs (DAGs) from neuronal networks

We obtain small-world models of the mouse visual cortex[5] and *C. Elegans* connectomes (Bock et al., 2011). *C. Elegans* neuronal network includes 2D spatial positions of the rostral ganglia neurons[6], while the mouse primary visual cortex was characterized with electron microscopy.

---

[4]Code for our experiments can be found at https://tinyurl.com/ofmicewormsandmen.

[5]https://neurodata.io/project/connectomes/, in graphml format.

[6]https://www.cise.ufl.edu/research/sparse/matrices/Newman/celegansneural.html

Table 1: Details of graphs used in our experiments. Small-world networks exhibit small average path lengths and high clustering coefficient.

|  | Watts-Strogatz | *C. Elegans* | Mouse Visual Cortex |
|---|---|---|---|
| Number of Nodes | 32 | 297 | 195 |
| Number of Edges | 64 | 2148 | 214 |
| Average Path length | 4.387 | 2.455 | 4.271 |
| Average Clustering Coefficient | 0.5 | 0.308 | 0.124 |
| Connected Components | 1 | 1 | 3 |
| Network Diameter | 8 | 5 | 8 |
| Average Degree | 4 | 14.465 | 1.097 |
| Modularity | 0.562 | 0.373 | 0.752 |

We treat the neuronal networks of both *C. Elegans* and the mouse visual cortex as undirected graphs which we convert to directed acyclic graphs (DAGs) in the same manner as (Xie et al., 2019) by randomly assigning vertices indices, and set the edge to point from the smaller index to the larger index, which enforces a partial ordering on vertices. We introduce source and sink nodes by connecting vertices with indegree 0 and outdegree 0 respectively to ensure singular input and singular output through the DAG graph. The source broadcasts copies to the input nodes, and the sink performs an unweighted average on output nodes.

## 3.2 Experiments

With the exception of our choices in graph topology and number of layers, we inherit our architecture from the "small regime" RandWire architecture described in (Xie et al., 2019), which includes two `conv` layers, followed by randomly wired modules, and a fully connected softmax layer. Our modifications to this include the use of one `conv` layer instead of two, and the use of a single 'random wiring' module as opposed to the three. Our single 'random wiring' module consists of one of the topologies described in the previous subsection.

As per the RandWire architecture, each node in the graph performs a weighted sum of the input, followed by a `ReLU-conv-BN` triplet transformation with a 3x3 separable convolution. Copies of the processed output are then broadcast to other nodes. We train for 50 epochs with Adam optimizer with learning rate of 0.001, batch size of 32 and with half-period cosine learning rate decay. For the first `conv` block before the DAG, we use a 2D `conv` with kernel size of 3, stride of 2 and padding of 1, followed by BN and ReLU.

We evaluated a model with one `conv` block and a fully connected layer without any DAG as a baseline. Furthermore, we evaluated the *C. Elegans* and mouse visual cortex DCNs and compared these with the best graph structure in (Xie et al., 2019), the Watts-Strogatz network. These were evaluated on MNIST, while the *C. Elegans* model was also evaluated on Fashion-MNIST and KMNIST.

# 4 Results and Analysis

In the generation of random graphs, Xie et. al. found that WS graphs performed better than ER and BA graphs. We thus compared DCNs with a simple convolutional graph-free CNN, and that with WS graphs as a baseline, and we observe that biologically wired DCNs outperform baselines without graphs, and with WS graphs.

Table 2: Performance on MNIST.

| Graph Model | Validation Accuracy (%) |
|---|---|
| Baseline, no graph | 98.01 |
| WS model, frozen | 97.05 |
| WS model, trainable | 99.27 |
| Mouse Visual Cortex, Frozen | 98.24 |
| Mouse Visual Cortex, Trainable | **99.30** |

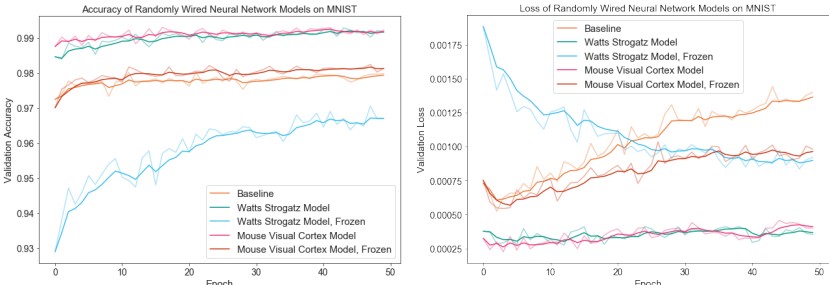

Figure 2: Validation accuracy (tested after each training epoch) and loss curves on MNIST. The frozen mouse visual cortex model achieves a higher validation accuracy than the baseline.

It could be argued that the accuracy improvement could be attributed to the increased number of parameters, so we further conduct experiments where we freeze the graph, thus keeping the same number of parameters as the graph-free CNN baseline. While the frozen WS model performs worse than the baseline, the Mouse visual cortex model performs better than the baseline even when frozen, suggesting that the graph topology is significant and independent of the number of parameters.

For the *C. Elegans* DCN, we evaluated the performance across different datasets and showed consistently competitive results on MNIST, KMNIST, and Fashion MNIST. Figure 3 shows the distribution of our results compiled from 10 training trials on each dataset. The mean test accuracy on MNIST was 99%, while we achieved 93% on KMNIST, and 90% on Fashion MNIST.

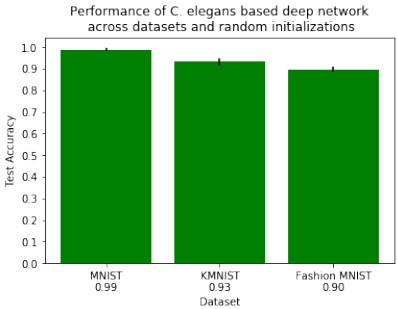

Figure 3: Distribution of accuracies of *C.Elegans* DCN on MNIST, KMNIST, and Fashion MNIST.

## 5   Conclusion

We demonstrated initial findings from applying networks inspired by real-world neuronal network topologies to deep learning. Our experiments show the trainability of a DNN based on the neuronal network *C.Elegans* and the visual cortex of a mouse with and without freezing the parameters of the graph modules, which outperforms WS graphs with good theoretical small-world properties.

In future work, we will examine more principled methods for constructing a DAG from these networks and examine the impact of spectral properties of the graph topologies used both in the architectures we proposed and in the architecture proposed by (Xie et al., 2019), while extending to other connectomes.

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
