# OpenReview forum: "Deep Connectomics Networks: Neural Network Architectures Inspired by Neuronal Networks"
_NeurIPS.cc/2019/Workshop/Neuro_AI — Real Neurons & Hidden Units @ NeurIPS 2019 Poster_

### Official Review · AnonReviewer3 · 2019-09-19
**Good concept not great execution**

**Clarity:** 3

**Category:**

Neuro->AI

**Clarity Comment:**

The introduction was well written however details were lacking in the methods and results. For example: "DAG" was never actually defined. Why was the c elegans network the only one tested on other datasets and not tested while frozen? Why does the validation accuracy start so high in Fig 2?

**Evaluation:**

3: Good

**Importance:**

4: Very important

**Importance Comment:**

Neuroscience certainly has seen an explosion in studies looking at network topology and applying the tools of graph theory/network science. Given how easily artificial neural networks can be mapped onto graph structures it seems very natural to combine the two. It is also a straightforward way to bring in biological data, potentially at exactly the right level of detail/abstraction. The particular results in this paper, however, do not reflect a particularly strong instantiation of this concept

**Intersection:**

4: High

**Intersection Comment:**

Bringing neuroanatomical data directly into deep nets is challenging and while many assumptions and simplifications were made in order to do it in this study, it is still an admirable attempt at combining neuroscience and AI.

**Rigor Comment:**

When claiming that a technique results in better performance on a task, the baseline network tested is obviously very important. The baseline model is described as containing one conv block and a fully connected layer. It seems that this baseline has far fewer processing stages and parameters than the models with DAGs. And the DAG models have differing numbers of units amongst themselves. Comparing to "frozen" (ie untrained) DAGs does not control for the benefit of having these extra nodes as even random weights can still perform well on simple tasks. Relatedly, the use of MNIST for the main comparison metric is a poor choice because the baseline model already performs so well that marginal increases are hard to interpret here. It seems that fashion mnist is a harder task at least and should have been used to compare models.

**Technical Rigor:**

2: Marginally convincing

---

### Official Review · AnonReviewer2 · 2019-09-26
**Not the best use of available data**

**Clarity:** 3

**Category:**

AI->Neuro

**Clarity Comment:**

The issue of number of trainable parameters is not sufficiently explored. I would have like to see a better exploration of how the results depend on this quantity and how things change with different assumptions about learned and unlearned connections.


**Evaluation:**

2: Poor

**Importance:**

2: Marginally important

**Importance Comment:**

The study attempts to use the wiring statistics of real brains to build neural networks. While it is an interesting approach, the choice of task and the model assumptions are not well suited to the topic. The performance improvements are also not very convincing, and it's unclear if we should expect these results to be generalizable.

**Intersection:**

3: Medium

**Intersection Comment:**

It's hard to believe that the C. elegans connectome would be optimized for MNIST in any way. Also, ignored directedness in the datasets is an unnecessary omission. More work could have been done to bring the models closer to the biology.

**Rigor Comment:**

The choice of network connectivity is poor. The authors use undirected networks and randomly convert them to directed networks, but connectome data with directed weights are readily available in a multitude of organisms, including C. elegans and mouse. The results are not convincing, with MNIST performance at above 97% in all cases. Why are results for C. elegans not shown in Table 2?

Additionally, the issue of number of trainable parameters is not explored. Freezing the weights is not sufficient -- more frozen parameters could still account for the performance benefits compared to the baseline.


**Technical Rigor:**

2: Marginally convincing

---

### Official Review · AnonReviewer1 · 2019-09-26
**Good concept, bad execution**

**Clarity:** 2

**Category:**

Neuro->AI

**Clarity Comment:**

The research is well-motivated, but the actual project pursued is not. The structure of the network models adopted and used is not clearly communicated to the reader (see technical rigor section) and the figures are lacking in detail. For example - half of the line plots in figure 2 (subfigures not individually labeled) are not described in either the text, the figure legend or the figure caption.

There is a clear message conveyed through this work, but it doesn't answer the questions presented in the ostensible thesis of the paper: how does network topology influence computation. They've shown that they can get high classification results on a particular sort of network architecture, but don't explore how the defining aspects of those topologies influence the results presented. The overall intent of the work is unclear for that reason

**Evaluation:**

2: Poor

**Importance:**

2: Marginally important

**Importance Comment:**

The connections between network topology and function in both neuroscience and AI research are very interesting. The pursuit of research at this intersection is highly important.

This paper does fall into that category work, but the methods and results presented therein do not add up to an important contribution to the area.

**Intersection:**

3: Medium

**Intersection Comment:**

Ideally, this would be highly intersectional; however, the lack of execution toward the stated intent of the paper do not follow through to actually fulfill that intersection. Understanding the role of network topology in network computation is important, but I think that the work presented here is less so.

**Rigor Comment:**

The paper goes to some length to motivate the research it presents, providing a brief survey of the development of network neuroscience that cites the connections between several prominent publications underlying that development. The technical aspects of their own work are detailed less satisfactorily.

The structure of the networks is presented in citation, but not actually detailed in any measurable way. Their method of constructing the networks is described in text reasonably well, but the diagrams presented (e.g. Fig 1) are not detailed nearly enough. It is not clear how the networks differ. Metrics are presented to describe the modular structure of the borrowed network subunits, but their connections to the desired topological results are not made clear.

Their results are also speciously presented. Four of the presented models start - without any prior training on the MNIST task - performed at >97% accuracy. Moreover, the results presented are explicitly labeled as validation performances. The loss patterns are also ill-detailed; increases in validation loss are not described and mesh strangely with the presented classification results.

**Technical Rigor:**

2: Marginally convincing

---

### Decision · Program_Chairs · 2019-10-02

Accept (Poster)